# Molecular Markers for Bladder Cancer Screening: An Insight into Bladder Cancer and FDA-Approved Biomarkers

**DOI:** 10.3390/ijms241814374

**Published:** 2023-09-21

**Authors:** Gabriela Vanessa Flores Monar, Thomas Reynolds, Maxie Gordon, David Moon, Chulso Moon

**Affiliations:** 1HJM Cancer Research Foundation Corporation, 10606 Candlewick Road, Lutherville, MD 2109, USA; 2NEXT Bio-Research Services, LLC, 11601 Ironbridge Road, Suite 101, Chester, VA 23831, USA; treynolds@nextmolecular.com; 3BCD Innovations USA, 10606 Candlewick Road, Lutherville, MD 2109, USA; 4Department of Otolaryngology-Head and Neck Surgery, The Johns Hopkins Medical Institution, Cancer Research Building II, 5M3, 1550 Orleans Street, Baltimore, MD 21205, USA

**Keywords:** bladder cancer, urinary biomarkers, cancer screening, surveillance

## Abstract

Bladder cancer is one of the most financially burdensome cancers globally, from its diagnostic to its terminal stages. The impact it imposes on patients and the medical community is substantial, exacerbated by the absence of disease-specific characteristics and limited disease-free spans. Frequent recurrences, impacting nearly half of the diagnosed population, require frequent and invasive monitoring. Given the advancing comprehension of its etiology and attributes, bladder cancer is an appealing candidate for screening strategies. Cystoscopy is the current gold standard for bladder cancer detection, but it is invasive and has the potential for undesired complications and elevated costs. Although urine cytology is a supplementary tool in select instances, its efficacy is limited due to its restricted sensitivity, mainly when targeting low-grade tumors. Although most of these assays exhibit higher sensitivity than urine cytology, clinical guidelines do not currently incorporate them. Consequently, it is necessary to explore novel screening assays to identify distinctive alterations exclusive to bladder cancer. Thus, integrating potential molecular assays requires further investigation through more extensive validation studies. Within this article, we offer a comprehensive overview of the critical features of bladder cancer while conducting a thorough analysis of the FDA-approved assays designed to diagnose and monitor its recurrences.

## 1. Introduction

In the global context, bladder cancer ranks as the 10th most prevalent cancer, witnessing 573,278 new cases and 213,000 reported deaths in 2020. Across different regions worldwide, there are notable disparities in the incidence rates of bladder cancer. Southern and Western Europe, as well as North America, stand out for their notably high rates of bladder cancer. Greece holds the distinction of having the highest incidence of bladder cancer among males globally, while Hungary leads in terms of incidence among females. Specifically, Southern Europe reports the most substantial bladder cancer incidence rates globally, with approximately 26.6 cases per 100,000 males and 5.8 cases per 100,000 females diagnosed each year. In contrast, regions such as Middle Africa, South Central Asia, and Western Africa, primarily consisting of countries with lower-than-average Human Development Index scores, tend to exhibit the lowest prevalence of bladder cancer [1]. Developed countries exhibit a higher incidence rate of bladder cancer, primarily affecting men, with incidence rates four times higher than in women. 

By 2023, it is estimated that the United States will experience around 62,420 new cases of bladder cancer in men (76%), ranking it as the fourth most prevalent cause of cancer among males. Over the last few decades, bladder cancer incidence has increased to the mid-2000s, followed by a decline of 1.8% per year from 2015 to 2019. However, this trend might vary among ethnicities or races [2]. In contrast to white men, both white women (hazard ratio [HR]: 1.20, 95% confidence interval [CI]: 1.17–1.23) and Black women (HR: 1.57, 95% CI: 1.49–1.66) exhibited an increased likelihood of bladder-cancer-related mortality, regardless of the stage, as indicated by the results [3].

Bladder cancer predominantly affects older individuals, with around 90% of diagnosed cases occurring in people over 55 years old. The average age for this cancer diagnosis is approximately 72 years for men and 75 years for women [4,5]. It also manifests considerable variations across different races. While the frequency of tumor incidence is twice as high among individuals of Caucasian descent compared to African Americans, the latter subgroup tends to experience a less favorable prognosis and a greater prevalence of advanced tumor stages upon presentation [6,7]. Pronounced disparities in mortality rates are observed among African Americans, older individuals, and female patients [8,9,10].

Despite a gradual decrease in incidence and prevalence, the medical costs associated with managing bladder cancer, including follow-up and complications, have significantly increased. In 2015, bladder cancer incurred direct and indirect medical costs of USD 7.93 billion in the U.S. This figure is anticipated to rise by 45% to reach USD 11.6 billion by 2030 [11]. Since follow-up and treatment for recurrences constitute 60% of the total medical costs, early detection of bladder cancer could potentially alleviate this economic burden [12].

While cystoscopy remains the established standard for bladder cancer detection, the approximate total cost per procedure, around USD 216.18 [13], coupled with its invasiveness and the potential for complications, significantly add to the overall cost burden. Voided urine cytology has long been used as a highly specific and non-invasive supplementary test compared to cystoscopy. However, it has two significant limitations: low sensitivity to detect low-grade tumors (ranging from only 4 to 31%) and dependence on the expertise of cytopathologists, leading to challenges in achieving consistent and high-quality readings [14].

These observations emphasize the urgency of incorporating new diagnostic tests in managing bladder cancer patients. The ideal characteristics of these new tests include being easy, better, faster, and cost-efficient for detecting bladder cancer, with a specific emphasis on monitoring low-grade papillary tumors. In addition, such non-invasive methods should rely on highly sensitive and specific bladder cancer markers to reduce the frequency of cystoscopies and improve the patient’s quality of life. Furthermore, enhancing the markers’ sensitivity in cases of high-grade disease is vital for the early identification of tumor recurrence and, consequently, for enhancing patient survival rates [15].

This article comprehensively overviews the current status and performance of FDA-approved urinary marker tests. Existing data suggest that some of these markers have the potential to play a role in the screening and surveillance of bladder cancer. Several new tests, especially for low-to-moderate grade stages, demonstrate significantly higher sensitivity than standard urine cytology and are commonly used. However, none have been accepted as a standard diagnostic modality within established clinical guidelines [16,17]. In order to establish the value of integrating markers into clinical decision making, well-designed protocols, and prospective, controlled trials need to be developed.

## 2. Bladder Cancer Overview

Bladder cancer is commonly identified as localized in the absence of metastatic disease. Then, it can be classified as non-muscle invasive bladder cancer (NMIBC) or muscle-invasive bladder cancer (MIBC). Approximately 70% of new bladder cancer cases are NMIBC, and this category is further categorized into low, intermediate, and high-risk groups after a transurethral resection or cystoscopy biopsy, according to the American Urologic Association’s risk classification [18]. 

The recurrence rate is high, with almost 60–80% of cases of NMIBC eventually recurring despite treatment [19]. Individuals diagnosed with low-risk and intermediate-risk NMIBC achieve 5-year recurrence-free survival rates of 43% and 33%, respectively. In contrast, among those with high-risk disease, as many as 21% will eventually progress to MIBC [20,21]. Based on histopathology and clinical presentations, two different types of NMIBC have been classified: the frequently recurring papillary tumor (Ta) and the more aggressive carcinoma in situ (CIS). While both types can progress into invasive tumors (T1–T4), the probability that low-grade Ta tumors progress to invasive disease is much less likely than high-grade Ta tumors and CIS [22,23]. In terms of histology, urothelial carcinoma makes up 75% of bladder cancer cases, while the remaining 25% is attributed to variant histology [24].

## 3. Risk Factors

### 3.1. Tobacco Consumption

Numerous environmental factors have been linked to bladder cancer. Among these, cigarette smoking stands out as the most established factor, contributing to approximately 55% of cases in the United States [25]. Tobacco consumption significantly raises the risk of developing bladder cancer by a factor of two to three [26].

A comprehensive meta-analysis encompassing 89 observational studies revealed that current smokers experience a threefold increase in bladder cancer risk in comparison to individuals who have never smoked (summary odds ratio [SOR] 3.14, 95% confidence interval [CI] 2.53–3.75) and twofold increased risk compared to former smokers (SOR 1.83, 95% CI 1.52–2.14). Even after 20 years post-smoking cessation, former smokers still show a 50% higher risk compared to those who have never smoked [27].

In addition, when examining 15 case–control studies, it becomes evident that a higher risk of urinary bladder cancer is associated more strongly with prolonged smoking over an extended period with a lower daily cigarette intake, rather than smoking a greater number of cigarettes per day for a shorter duration when considering equal pack years [28].

In a meta-analysis comprising 17 studies, it was determined that active smokers exhibited a higher risk of mortality after radical cystectomy with a hazard ratio [HR] of 1.21 and a 95% confidence interval [CI] of 1.08–1.36, *p* = 0.001, a greater likelihood of cancer-specific mortality (HR 1.24, 95% CI 1.13–1.36, *p* < 0.001), and an elevated risk of bladder cancer recurrence (HR 1.24, 95% CI 1.12–1.38, *p* < 0.001) [29]. Luckily, in 2014, the Centers for Disease Control and Prevention (CDC) showed a report of the significant fall in smoking trends in the U.S. among adults from 42.4% in 1965 to 16.8% [30].

### 3.2. Occupational Exposure

When considering occupation, the greatest risks of bladder cancer were associated with jobs involving exposure to aromatic amines (such as tobacco, dye, rubber industry workers, hairdressers, printers, and leather workers) and polycyclic aromatic hydrocarbons (including chimney sweeps, nurses, waitstaff, aluminum workers, seamen, and oil/petroleum industry workers) [31].

Occupational exposures to these diverse carcinogens including aromatic amines, toluene, perchloroethylene, polycyclic aromatic hydrocarbons (PAHs), and metalworking fluids are responsible for around 20% of cases [32,33,34]. A systematic review and meta-analysis involving 263 studies showed that individuals exposed to aromatic amines exhibited the highest incidence of bladder cancer. Conversely, occupations characterized by exposures to PAHs and heavy metals are associated with the highest risks of bladder-cancer-related mortality [35]. Noteworthy occupational sectors with the most significant bladder cancer susceptibility include those within the paint, dye, rubber, metal, and petroleum industries. Furthermore, emerging data indicate an elevated bladder cancer incidence among firefighters [31], attributed to their exposure to combustion byproducts such as PAHs and benzene [36,37]. Moreover, individuals with substantial occupational exposure to diesel exhaust fumes, such as bus and truck drivers, railroad workers, and heavy equipment engine mechanics, also demonstrate an increased risk of bladder cancer [38].

In a population-based study conducted by Rushton et al., it was determined that 7.1% of bladder cancer cases in men could be attributed to occupational factors, while no such attribution was observed in women [35]. Furthermore, a case–control study revealed statistically significant elevated risk in men employed as machine operators in the printing industry, while male farmers exhibited a reduced risk. Among women, after accounting for smoking duration, no notable associations between occupation and bladder cancer risk were identified (HR: 5.4; 96% CI, 1.6–17.7) [32].

### 3.3. Cardiovascular Disease

A retrospective analysis, conducted across multiple institutions, involved a substantial population of over 2000 patients who underwent trans-urethral resection of the bladder. Among the participants, more than 81% received a confirmed diagnosis of bladder cancer through pathological evaluation. The study highlighted that cardiovascular disease acted as an independent protective factor against bladder cancer, but this effect was not observed in cases of high-risk tumors [39]. It is widely recognized that low-risk and high-risk cancers follow distinct pathways. In low-risk cases, altered cells typically progress through hyperplasia toward the development of low-grade tumors. In contrast, in high-risk cases, these cells become dysplastic, often involving the tp53 mutation, and follow the CIS pathway, which can eventually lead to invasive carcinoma [40]. This mechanism should also be considered to gain a deeper insight into why cardiovascular disease fails to exert its protective effect on high-risk tumor development [39]. 

### 3.4. Genetic Susceptibility

Currently, no widely recognized genetic or hereditary basis has been established for bladder cancer. Nevertheless, emerging research suggests that genomic instability and mutations or alterations in genetic pathways may contribute to the development of bladder cancer. Some studies indicate that specific gene variations in GSTM-1 and NAT-2, which are involved in detoxifying carcinogens, could potentially increase the susceptibility of certain individuals to bladder cancer [41].

Factors like slow acetylation may not inherently result in bladder cancer but could increase susceptibility to carcinogens, such as those found in tobacco products. N-acetyl transferase enzymes (NAT1, NAT2) play a role in both activating and detoxifying these carcinogens. Notably, individuals with a slow NAT2 acetylator genotype were identified as a significant risk group for bladder cancer, particularly among smokers (HR: 1.31; 95% CI, 1.01–1.70) [42].

Genetically speaking, single-nucleotide polymorphisms (SNPs), within specific genes situated on chromosome 8q24, with a focus on the PSCA gene, have been associated with a notably higher risk of bladder cancer (OR 1.33). The PSCA gene features an androgen response element (ARE) in its promoter region, leading to the suggestion that this region may lose its responsiveness to androgen receptors (ARs). Consequently, it could develop an independent mechanism not reliant on androgens, which increases the potential for metastasis. To illustrate this, a reduction in the affinity of AR binding to the ARE within the PSCA gene due to such an SNP might initiate pathways independent of androgens, such as IGFBP2, which, in turn, would foster tumor growth and metastatic spread. It could be postulated that alterations in androgen levels in females might accelerate the activation of this mechanism, potentially contributing to the more aggressive tumor behavior noted in women with bladder cancer [43].

### 3.5. Physical Activity

In an innovative approach, one study stands out as being one of the first investigations of its kind. It conducted a comprehensive prospective cohort spanning over two decades, involving more than 2000 newly diagnosed bladder cancer cases. It showed a connection between prolonged periods of sitting and the incidence of invasive bladder cancer. Specifically, individuals who engaged in 6 or more hours of daily sitting exhibited a 22% higher risk of developing invasive bladder cancer compared to those who spent less than 3 h seated each day. Notably, this correlation retained its statistical significance even after adjustments for critical risk factors such as smoking status, moderate-to-vigorous physical activity (MVPA), and body mass index (BMI) [44].

### 3.6. Consumption of Red Processed Meat

Numerous epidemiological investigations have explored the relationship between the consumption of red or processed meats and the incidence of bladder cancer. These studies have revealed a direct link between bladder cancer risk and the consumption of processed meats, which undergo processes such as salting, fermentation, smoking, or other treatments [45,46]. Specifically, a 20% increase in bladder cancer risk is associated with a daily intake of 50 g of processed meat (RR = 1.20, 95% CI: 1.06–1.37) [47]. A prospective study also identified a positive correlation between the consumption of processed red meat and bladder cancer risk, even after adjusting for potential confounding factors (HR = 1.47, 95% CI: 1.12–1.93) [48].

### 3.7. Gender

In terms of physiology, the normal urothelium in the bladder expresses both androgen receptors and estrogen receptors alpha and beta (ERa and ERb) [49,50]. Interestingly, while testosterone appears to promote the onset of bladder cancer, exposure to estrogen may initially offer protection against bladder cancer development, potentially contributing to the fact that women are nearly three times less likely to be diagnosed with bladder cancer than men. However, once bladder cancer is established, estrogens may play a role in promoting its progression [51,52].

Although the association between the expression of androgen receptors and the stage and grade of bladder cancer is a topic of debate, most studies indicate a decreased detection of androgen receptors in cases of high-grade and high-stage disease [53]. For instance, Miyamoto et al. [54] demonstrated a notably lower presence of androgen receptors in high-grade and muscle-invasive bladder cancer compared to low-grade bladder cancer (*p* = 0.023) and NMIBC (*p* = 0.018), respectively. In line with these findings, another research group reported that the expression of androgen receptors is lower in T2 tumors (21%) when compared to Ta (60%) and T1 (60%) tumors [55]. Additionally, a study examining the mRNA expression of androgen receptors in bladder cancer cell lines disclosed an inverse relationship between the transcript expression of androgen receptors and the severity, stage, and spread of bladder cancer [56,57].

As mentioned, the incidence of bladder cancer is clearly higher in men than in women. However, when evaluating the ratio of cancer-specific mortality (CSM) to the incidence of bladder cancer, it becomes evident that women face a greater risk of CSM in bladder cancer [1]. Furthermore, women tend to receive diagnoses of locally advanced disease more frequently and have a higher proportion of nonurothelial cell types at the time of diagnosis compared to men [58].

In cases where patients presented with hematuria and were later diagnosed with bladder cancer, women experienced a significantly longer period from their initial hematuria report to the bladder cancer diagnosis compared to men (85.4 vs. 73.6 days; *p* < 0.001). Additionally, women presenting with hematuria were more likely than men to be diagnosed with a urinary tract infection (OR 2.32) and less likely to undergo abdominal or pelvic imaging (OR 0.80) [59]. 

A research study [60] noted that there was no significant gender-based differences in clinical symptoms at the time of initial presentation, including hematuria and irritative lower urinary tract symptoms, among patients newly diagnosed with bladder cancer. However, a substantial gender discrepancy arose in healthcare-seeking behaviors: 78% of men, as opposed to 55% of women, consulted with a urologist (*p* < 0.05). Moreover, prior to their bladder cancer diagnosis, symptomatic treatment without further diagnostic evaluation was administered to 19% of men, whereas this proportion significantly increased to 47% for women (*p* < 0.05). Alarmingly, 16% of women received three or more courses of treatment for presumed urinary tract infections. Importantly, it was consequently found that patients presenting with hematuria and given a delayed diagnosis for bladder cancer faced a considerably elevated risk of cancer-specific mortality [61].

## 4. Presentation and Diagnosis

The predominant symptom commonly encountered upon initial patient presentation is painless hematuria, which may manifest either as gross or microscopic hematuria [18]. The existence of over three red blood cells per high-power field characterizes microhematuria [62]. It is considered to be one of the most common reasons for urology evaluations, with prevalence rates ranging from 2.4% to 31.1% among healthy individuals in screening studies. Approximately 3% of patients with microhematuria have genitourinary malignancy [63,64]. To avoid unnecessary evaluation and treatment while presenting microhematuria, the AUA and SUFU established a stratification of risks of patients into low, intermediate, and high-risk categories for genitourinary malignancy within their guidelines, allowing for a more individualized approach to follow-up and treatment [7].

## 5. Urine Cytology

Urine cytology has been the established non-invasive method for the detection and follow-up of bladder cancer in combination with cystoscopy. As mentioned above, urine cytology exhibits high specificity but suffers from a significant weakness in its very low sensitivity to well-differentiated bladder cancer [65]. Cytology serves as a valuable tool, especially when used alongside cystoscopy, for identifying high-grade (G3) tumors, but is not intended for the detection of low-grade tumors. A meta-analysis reported sensitivity of 34% and specificity of 99% [66]. It is essential to consider several factors responsible for this poor sensitivity. First, only a small volume of voided urine can be processed, and the limited amount of urine samples makes it challenging to identify tumor cells among other cells like red blood cells (RBCs) and leukocytes. Second, there are no clear objective criteria that differentiate between low-grade tumors and reactive cells, leading to potential confusion and interobserver variation [67].

## 6. Cystoscopy

The gold standard for diagnosing bladder cancer in cases of hematuria has been the visual inspection of the bladder through cystoscopy. Most guidelines recommend cystoscopy when hematuria is visibly present [16,18]. Conventional white light cystoscopy outperforms imaging methods, achieving sensitivities ranging from 87% to 100% and specificities ranging from 64% to 100%. When cystoscopy identifies a suspicious lesion, transurethral resection is conducted to either confirm the presence of bladder cancer (true positive) or rule it out (false positive) [68]. However, in cases of non-visible hematuria, ongoing debates revolve around the necessity of diagnostic cystoscopy, given the lower incidence of bladder cancer in this subgroup [69,70].

In patients with NMIBC, enhanced cystoscopy should be offered at the time of TURBT if available to improve tumor detection and reduce the likelihood of recurrence [18]. These enhanced cystoscopy techniques, such as narrow-band imaging and blue light cystoscopy, increase the accuracy in identifying bladder tumors during both diagnostic cystoscopy and endoscopic resection. Narrow-band imaging, for instance, increases the detection rate by approximately 10% on a per-patient basis and 20% on a per-lesion basis, while reducing the risk of recurrence at 3 and 12 months by 34% [71]. 

Blue light cystoscopy detects as much as 14% of papillary Ta/T1 lesions and 40% of CIS lesions that are often missed using conventional cystoscopy [72]. A large retrospective study by Todenhöfer et al. [73] showed that despite the initially higher median costs associated with blue light cystoscopy compared to white light cystoscopy, these expenses are balanced out by reduced average follow-up costs in the long run due to enhanced patient outcomes in the blue light cohort group.

## 7. FDA-Approved Urinary Markers for Screening and Surveillance of Bladder Cancer

Two key aspects mean that bladder cancer screening will be important in the upcoming decades. First, the ongoing contribution of smoking as a critical hazard to long-term carcinogenic effects. Second, bladder cancer is highly unlikely to metastasize before becoming invasive [23]. Therefore, there is a valuable opportunity for the early detection of bladder cancer within the time window between tumor origination and invasion. The management of non-invasive cancers is associated with fewer morbidities and is more effective than that of invasive tumors [74], as at this stage of tumor development, cystectomy, systemic chemotherapy, or chemo-radiation therapy are not required.

Currently, six urinary biomarker tests are approved for the diagnosis or surveillance of bladder cancer (Table 1): quantitative nuclear matrix protein 22 (NMP22) (Alere NMP22), qualitative NMP22 (BladderChek), qualitative bladder tumor antigen (BTA) (BTA stat), quantitative BTA (BTA TRAK), fluorescence in situ hybridization (FISH) (UroVysion), and fluorescent immunohistochemistry (ImmunoCyt). The qualitative NMP22 and BTA tests can be performed as point-of-care tests, while the others must be conducted in a laboratory [75].

### 7.1. Bladder Tumor Antigen Assay: BTA Test

The BTA Stat and BTA TRAK tests are two in vitro immunoassays that detect human complement factor H-related protein (hCFHrp) in the urine samples of patients with urothelial carcinoma [76]. A high concentration of hCFHrp in the urine of patients with bladder cancer hinders the easy detection of hCFH, which is present in minimal quantities in normal healthy individuals. Therefore, the accuracy of the tests relies on the detection of hCFHrp, not hCFH [77,78,79,80]. BTA Stat is a qualitative point-of-care assay that provides results within five minutes and requires basic training. On the other hand, BTA TRAK is a quantitative ELISA that requires dedicated and trained personnel and is performed in a designated laboratory, taking several hours to produce the final reports. BTA TRAK utilizes an anti-hCFHrp monoclonal antibody, enabling the quantitative detection of the target antigen in urine [81].

These two tests have been approved by the FDA only for monitoring recurrences in subjects with a history of bladder cancer in conjunction with cystoscopy [82]. Although BTA assays initially sparked excitement, their use has considerably declined in the last ten years due to low specificity, a high number of false positive cases, and increased regulatory controls with declining reimbursement by Medicare and private health insurance companies [83].

Subset analysis of recurrent tumor stratified by grade showed lower sensitivities for grade 1 and 2 tumors for both BTA stat and TRAK (grade 1 = 45 and 55%, respectively; grade 2 = 60 and 59%, respectively) as compared to grade 3 tumors (75 and 74%, respectively). A trend of increasing sensitivity and specificity for overall tumor detection was noted with increasing tumor stages [84]. Furthermore, the BTA stat test has been shown to have lower sensitivity for detecting recurrent as opposed to primary tumors; possibly related to the smaller size of recurrent tumors, BTA TRAK showed increasing sensitivity and specificity with higher tumor grades and stages [85].

In a meta-analysis of BTA Stat comprising 13 studies, including 3462 patients, the test demonstrated sensitivity of 67% (95% confidence interval 64–69%), higher than the sensitivity of urine cytology of 43% (95% confidence interval 40–46%). However, its specificity was inferior to that of cytology. BTA Stat had higher sensitivity for high-grade tumors (74%) than for low-grade tumors (25%), with specificity of 77% [86]. 

Chou et al. performed a meta-analysis examining the various sensitivities and specificities of FDA-approved urinary biomarkers, as presented in Table 2 and Table 3. Qualitative BTA exhibited sensitivities ranging from 55% to 83%, with average sensitivity of 64%, and specificities varying from 66% to 87%, with average specificity of 77%. In the case of quantitative BTA, sensitivities ranged from 46% to 87%, with average sensitivity of 65%, and specificities ranged from 38% to 85%, with an overall average of 74% [87].

Although they have higher sensitivity than cytology, these tests still present high false positive rates, most commonly due to blood in the urine samples tested. Complement factor H is usually present in the blood, leading to inevitable false positive BTA Stat or TRAK tests when there is hematuria [88]. It is essential to consider certain conditions responsible for high false positives, such as infection, hematuria, dysuria, incontinence, a history of intravesical therapy, ureteral stents or nephrostomy tubes, renal or bladder calculi, benign inflammatory disease, intestinal interpositions, or other genitourinary cancers, which are also responsible for most of the false positives observed in almost all molecular tests for bladder cancer [82]. False positives for up to 2 years after intravesical bacillus Calmette–Guerin (BCG) therapy limit the usefulness of BTA tests in the monitoring of recurrent tumors [80]. Compared to the BTA TRAK test, false positives are frequently seen with BTA Stat. False positives are seen in <5% of subjects with no known urinary pathology. Specific exclusion criteria can improve the performance of both BTA tests, although distinguishing between the clinical presentations of benign inflammatory conditions and urothelial carcinoma can sometimes be challenging [82].

### 7.2. NMP22

Nuclear matrix proteins (NMPs) constitute a group of proteins that play a significant role in the structural framework of the nucleus, providing support and participating in various processes, from DNA replication to the regulation of gene expression. Numerous NMPs are overexpressed in urothelial tumors and can be found in the urine after tumor cell apoptosis. NMP22 stands out as the protein that has been the most extensively investigated and has been used to diagnose bladder cancer and monitor its recurrence. Both the NMP22 Bladder Cancer ELISA, a quantitative test, and the NMP22 BladderChek, a point-of-care test, have received FDA approval for surveillance [76]. However, only BladderCheck has received approval for the initial diagnosis, specifically in symptomatic patients or individuals at an increased risk of developing bladder cancer [89].

In 2015, Chou et al. conducted a meta-analysis, demonstrating sensitivity of 69% and specificity of 77% for the quantitative NMP22 ELISA test. The corresponding value for the point-of-care test was 58% for sensitivity and 88% for specificity [87]. 

Wang et al. conducted a separate meta-analysis of 19 studies using the point-of-care test NMP22, which included 5291 patients. It demonstrated sensitivity ranging from 52% to 59% and specificity ranging from 87% to 89% [90]. It was also revealed that the sensitivity of the NMP22 test varied depending on the tumor stage and grade. Specifically, when considering Ta tumors, the sensitivity of the test was relatively low. However, as the tumor stage increased, moving from Ta to T1 and >T2, the sensitivity of the test demonstrated a steady rise, with rates of 13.68%, 29.49%, and 74.03%, respectively. A similar ascending trend was observed concerning tumor grade, with sensitivities of 44.16% for G1, 56.25% for G2, and 74.03% for G3. When the results from multiple studies were pooled together, the NMP22 test exhibited better potential for detecting >T2 stage tumors and high-grade bladder cancer. Additionally, subgroup analysis indicated that the test performed more effectively in detecting bladder cancer in Asian populations compared to Caucasian populations [90,91].

Diverse threshold values (ranging from 3.6 to 12 U/mL) have been employed; however, reducing the threshold may augment sensitivity at the expense of specificity. In particular, the assay’s sensitivity is markedly diminished for T1 and non-invasive lesions (ranging from 42% to 76%) compared to the one in muscle-invasive tumors. The increased sensitivity compared to urinary cytology manifests itself primarily in identifying low-grade and low-stage bladder cancers [92]. NMP22 and other urinary bladder markers exhibit higher performance in patients with disease advanced stage and with increased biological aggressiveness [93,94]. For example, in an evaluation by Poulakis et al. [95], involving 739 patients and utilizing an NMP22 cutoff of ≥8.25 U/mL, sensitivities of 79% (165/208), 90% (83/92), and 97% (96/99) were observed in patients with 1, 2–3, and >3 tumors, respectively. Meanwhile, a study by Sanchez Carbayo et al. [96] with 187 patients utilizing an NMP22 cutoff of ≥14.6 U/mL revealed sensitivities of 72% (18/25) and 75% (61/81) in patients with single and multiple tumors, respectively. This variability could be attributed to the level of NMP22 that reaches the threshold, depending on the amount of apoptotic cell debris excreted into the urine (the basis for a positive test).

In a study by Sharma et al. [97], which included 287 symptomatic patients, NMP22 and BTA stat were evaluated. The findings revealed that more than 80% of false positive results were linked to various clinical conditions, including benign inflammatory or infectious states, renal or bladder calculi, recent history of foreign objects within the urinary tract, bowel interposition segments, alternate genitourinary malignancies, or samples acquired through urinary instrumentation. The study subsequently excluded cases falling within the aforementioned clinical categories, demonstrating enhanced specificity and positive predictive value (PPV) for NMP22 (95.6%, 87.5%) and BTA stat (91.5%, 69.7%). 

An important aspect to consider in interpreting the NMP22 test results is the potential occurrence of a positive result from a urine-based marker before the visualization of an actual tumor. As discussed later in this context, anticipatory positive outcomes have also been documented with the NMP22 test [88,98]; however, this phenomenon appears to be less frequent compared to other tests. Numerous investigations have indicated a higher probability of clinical recurrence in patients who show a positive fluorescence in situ hybridization (FISH) assay, in contrast to those who demonstrate negative assays in negative cystoscopy-guided tumor detection [97,99,100,101,102,103].

### 7.3. UroVysion^®^

UroVysion constitutes a multi-chromosomal fluorescence in situ hybridization (FISH) assay designed to identify aneuploidy involving chromosomes 3, 7, or 17 and the loss of the 9p21 locus. The FDA authorized the utilization of this assay to diagnose and monitor instances of urothelial carcinoma. The established criteria for detecting bladder cancer through this test require at least one of the following conditions: a. A minimum of four cells demonstrating gains of at least two chromosomes within the same cell (out of 25 cells). b. Ten or more cells displaying a gain of a single chromosome. c. Ten or more cells showcasing tetrasomic signal patterns. d. Over 20% of cells exhibiting a loss of the 9p21 locus [82].

The sensitivity of UroVysion ranges from 69% to 87%, and its specificity ranges from 89% to 96% [104,105]. UroVysion has demonstrated excellent sensitivity in detecting carcinoma in situ and high-grade tumors, with sensitivities ranging between 83% and 100% [82]. Moreover, it serves as a valuable adjunct to cytology as it maintains the specificity of cytology while simultaneously increasing sensitivity (45.8% vs. 72.2%) [102,106]. One of the key advantages of this test is its high specificity, as it remains unaffected by hematuria, inflammation, and other conditions that may cause false positive readings with some other tumor markers. Additionally, UroVysion has shown potential for monitoring patients with non-muscle-invasive bladder cancer to assess their response to intravesical therapy [107].

In earlier case–control studies [108], the sensitivity of FISH varied between 69% and 87%. These studies consistently reported the low sensitivity of FISH for low-grade (36–57%) and low-stage (62–65%) tumors, while it demonstrated high sensitivity for high-grade and high-stage tumors (83–97%). Initially, the detection of carcinoma in situ was reported to be close to 100%. However, the limited performance of FISH in low-grade or low-stage tumors has been inconsistent across all studies. Several reports have highlighted the superiority of FISH over cytology. 

UroVysion^®^ is more sensitive for carcinoma in situ (approximately 100%) and high-grade tumors (83–97%) than for low-grade tumors (36–57%), as the target abnormalities are often absent from low-grade lesions. One apparent advantage of UroVysion is the detection of occult tumors that are not initially visible using cystoscopy. Chromosomal abnormalities detected in exfoliated cells have been shown to precede cystoscopically identifiable urothelial carcinoma by 0.25–1 year in 41–89% of patients under surveillance [88,101,102,103].

A study involving 1835 paired urine samples assessed the effectiveness of UroVysion and cytology for detecting bladder cancer. Of these samples, 1045 were obtained from patients undergoing recurrent UCC surveillance, while 790 were collected due to hematuria. When it came to detecting UCC, the combined results revealed that FISH achieved overall sensitivity of 61.9%, specificity of 89.7%, a positive predictive value of 53.9%, and a negative predictive value of 92.4%. In contrast, cytology had overall sensitivity of 29.1%, specificity of 96.9%, a positive predictive value of 64.4%, and a negative predictive value of 87.5%. Both FISH and cytology performed better in the surveillance population and in samples with high-grade UCC. Notably, among 296 cases with atypical cytology that were confirmed to have UCC, 61 cases, mostly featuring high-grade UCC, tested positive using the multiprobe FISH assay [109].

A comprehensive analysis of several studies has yielded a reported sensitivity of 0.63 (95% confidence interval [CI], 0.50 to 0.75) and specificity of 0.87 (95% CI, 0.79 to 0.93) for fluorescence in situ hybridization (FISH) in 11 investigations. When employed for surveillance purposes, FISH exhibited sensitivity of 0.55 (95% CI, 0.36 to 0.72) and specificity of 0.80 (95% CI, 0.66 to 0.89). To assess symptoms, the sensitivity of FISH was determined to be 0.73 (95% CI, 0.50 to 0.88) [110,111]. The broad spectrum of sensitivity and specificity values reported for UroVysion FISH across various studies likely stems from variations in patient selection, study design, tumor prevalence, and technical discrepancies between testing laboratories. In specific meta-analyses, sensitivity has been observed to surpass 70%, and even approach 80%, notably when excluding small and low-grade lesions from the data analysis. Concerning specificity, a broad range from 43% to 100% has been documented [112].

While the fluorescence in situ hybridization (FISH) test has shown a relatively elevated rate of false positive outcomes, multiple studies have suggested that the decreased specificity observed in subsequent surveillance tests could be attributed to an “anticipatory positive result”. This phenomenon entails detecting premalignant changes using the FISH test preceding the identification of recurrent disease through cystoscopy [111,112,113,114,115,116,117,118,119,120,121,122,123]. For instance, a study [122] revealed that 89% of individuals with a false positive FISH test exhibited a positive bladder biopsy within a year of the test. At the same time, another investigation found that FISH preceded tumor recurrence in 85% of cases. Nevertheless, the precise significance of an anticipatory positive result remains unclear, given that many non-muscle-invasive bladder cancer patients eventually experience disease recurrence [123].

In a recent development, Gopalakrishna et al. conducted an extensive review encompassing all Duke University Medical Center USA patients who underwent urine cytology or UroVysion FISH alongside cystoscopy between 2003 and 2012. This comprehensive study involved 6729 urine tests (4729 cytology and 2040 UroVysion FISH) which were matched with cystoscopies, serving as the gold standard. The sensitivity and specificity values obtained were 63% and 41% for cytology and 37% and 84% for UroVysion FISH, respectively. A one-year lag time was considered for cancer anticipation, yet neither test demonstrated improved anticipatory results. In patients with positive cytology and initially negative cystoscopy, the hazard ratio for developing a bladder tumor within one year was 1.83; 76% experienced tumor development within the year. Similarly, among individuals with a positive FISH result and initially negative cystoscopy, the hazard ratio for developing a bladder tumor in one year was 1.56; 40% of these patients developed a tumor within one year. This study, representing one of the largest-scale endeavors conducted recently, demonstrated lower sensitivity for the FISH test, and no conclusive anticipatory role from FISH was established [124].

Regarding the prognostic role of FISH, Ng et al. [76] demonstrated its predictive significance among NMIBC patients exhibiting negative cystoscopy results and suspicious cytology. Positive FISH results emerged as a substantial predictor of recurrence (hazard ratio [H.R.]: 2.35; 95% confidence interval [CI]: 1.42–3.90, *p* = 0.001) in multivariable analysis and progression (H.R.: 3.01; 95% CI: 1.10–8.21, *p* = 0.03) in univariable analysis. Moreover, it has been highlighted that opting to omit bladder biopsy due to negative UroVysion findings in cases of atypical cytology and negative or inconclusive cystoscopy outcomes has exhibited cost-effectiveness and the potential to reduce unnecessary adverse events [77].

### 7.4. ImmunoCyt

The ImmunoCyt test is an immunocytofluorescence-based test that detects carcinoembryonic antigens and sulfated mucin glycoproteins expressed on most B.C. cells but not normal cells, using fluorescently labeled monoclonal antibodies. The sensitivity of this assay varies widely among studies, ranging from 60% to 100%, with specificity of 75%−84% [124,125,126].

A minimum of 500 cells is required for evaluation, and a single fluorescent cell is regarded as a positive test. ImmunoCyt is an FDA-approved test indicated as “an aid” to urothelial carcinoma management in conjunction with urine cytology and cystoscopy. Initial analysis showed that this test possesses median sensitivity and specificity of 81% and 75%, respectively. Some investigators have suggested that sensitivity is more significant for high-grade tumors, whereas others report comparable sensitivity across all tumor stages and grades [127,128].

In a meta-analysis of seven studies, ImmunoCyt demonstrated superior pooled sensitivity of 72.5% when compared to urine cytology, which yielded sensitivity of 56.6%. However, ImmunoCyt exhibited lower specificity, measuring 65.7% as opposed to cytology’s specificity of 90.6% [129]. This assay is less susceptible to the influence of hematuria and inflammation compared to other diagnostic methods, although it can be affected by the presence of urinary tract infections, urolithiasis, and benign prostatic hyperplasia [130]. Moreover, the technology exhibits significant variability in interpretations between different observers and necessitates the involvement of cytopathologists to ensure accurate implementation. Consequently, its adoption in clinical practice has been restricted [131,132].

### 7.5. Synthesizing Screening Recommendations

An ideal marker should exhibit high overall sensitivity, detecting both low- and high-grade diseases while maintaining high specificity and being cost-effective with point-of-care capabilities. In a screening scenario, the emphasis is on high specificity, as a test with low specificity could lead to an unacceptably high number of patients undergoing further evaluation [133]. On the other hand, in follow-up settings, sensitivity becomes more crucial to ensure that bladder cancer persistence or recurrence is not missed. Additionally, the diagnostic power of the assay may also influence marker selection, with some urologists favoring markers with good performance in low-grade disease, while others prioritize high-grade tumor detection. Since approximately 70% of all bladder tumors are low-grade, the former group seeks to detect these early-stage tumors as soon as possible. In contrast, the latter group focuses on the early detection of high-grade tumors, which pose a higher risk and require aggressive treatment [134].

As discussed, most molecular detection tests for urothelial carcinoma have median sensitivities below 90%. One approach to increase sensitivity is to use a panel of molecular markers or combine markers with cytology or cystoscopy [135]. For example, Têtu et al. reported that while combining cytology and ImmunoCyt increased sensitivity for detecting recurrent urothelial carcinoma, it also reduced the respective specificities of the individual tests, resulting in specificity of 61% for the combination [128].

In response to the suboptimal performances of individual biomarkers, efforts have turned toward utilizing marker panels to enhance sensitivity. Another research study discovered that combining two tests, including cytology, immunocytology, FISH, and NMP22, yielded sensitivity and negative predictive values not exceeding 89.8% (Immunocyt + NMP22) and 92.1% (FISH + Immunocyt) [136]. In scenarios where cytology is supplemented with any of the four tests, the corresponding values do not surpass 86.7% (NMP22) and 91.3% (immunocytology). The addition of FISH to traditional urine cytology is correlated with sensitivity of 80.5% (94.0% for high-risk tumors) and a negative predictive value of 90.1% (98.8% for high-risk tumors) [137].

### 7.6. BCG Treatment and Reflex Testing

According to the AUA/SUO Guideline, clinicians have the option to make use of biomarkers like UroVysion FISH and ImmunoCyt for assessing responses to intravesical BCG therapy and addressing equivocal cytology outcomes [18,19]. A critical study involving sequential UroVysion FISH assessments in patients undergoing BCG treatment demonstrated clinical significance. Specifically, abnormal test outcomes at baseline (prior to BCG initiation), at the 6-week mark (before the sixth BCG instillation), and prior to the 3-month cystoscopy (before the first maintenance course) were notably linked to both cancer recurrence and progression. In the third month, UroVysion^®^ FISH successfully identified 50% of patients who encountered cancer progression within two years, with half of these individuals yielding a positive test outcome compared to only 3% among those with a normal test result [138].

An alternative approach to enhance surveillance protocols for NMIBC and optimize costs involves using reflex testing. Incorporating subsequent highly sensitive biomarkers in patients with negative or uncertain results from initial tests significantly improves the accuracy of follow-up assessments. Notably, the presence of immunotherapy-induced inflammatory changes in the bladder can complicate reliable evaluations of the lower urinary tract, thereby limiting the effectiveness of cytology as an adjunct to cystoscopy for detecting carcinoma in situ (CIS) or upper tract lesions during BCG treatment. To address this challenge, the performance of FISH (fluorescence in situ hybridization) and ImmunoCyt was explored in individuals with atypical cytology. UroVysion FISH exhibited 100% sensitivity and 100% negative predictive value in patients with negative cystoscopy yet equivocal cytology [65]. On the other hand, ImmunoCyt demonstrated 73% sensitivity in detecting recurrent bladder tumors in patients with atypical cytology, alongside an associated negative predictive value of 80% [139]. In conclusion, both FISH and ImmunoCyt are acknowledged by the American Urological Association (AUA)/Society of Urologic Oncology (SUO) as potential reflex biomarkers to assist in adjudicating atypical cytology and thereby reducing the need for unnecessary workups [140].

### 7.7. Recurrence Detection in Cases of Negative Cystoscopy

For individuals who have negative cystoscopy results, biomarkers could play a role in monitoring the recurrence of NMIBC. In this case, cytology displayed an HR of 3.9 (with a 95% confidence interval [CI] of 1.75–9.2, *p* < 0.001), UroVysion demonstrated an HR of 6.2 (95% CI: 1.7–29.7, *p* = 0.004), uCyt+ showed an HR of 5.1 (95% CI: 1.4–23.8, *p* = 0.01), and NMP22-ELISA presented an HR of 2.4 (95% CI: 0.7–11.1, *p* = 0.19). In the group of patients with negative cytology results, only the NMP22-ELISA test was associated with an increased risk of recurrence when it provided a positive result (*p* = 0.01). Interestingly, when both cytology and the NMP22 test were negative, only 13.5% of patients experienced relapse, and 5.4% progressed after a 24-month period [141].

### 7.8. Discussing Biomarker Use with the Patient

Considering the relatively low incidence of bladder cancer in both the general population (0.001%) and individuals aged 50 years and above (0.67–1.13%), implementing mass screening for this condition would not be economically viable due to the likelihood of identifying a considerable number of false positive cases that necessitate unnecessary further investigations [142]. Consequently, research concerning bladder cancer screening has primarily focused on specific high-risk populations to optimize the utility of such screening interventions [61].

Discussing the screening algorithm with the patient should include a biomarker test with reasonable sensitivity and cost-effectiveness for bladder cancer screening. Providers should possess extensive expertise in the screening modalities utilized for bladder cancer and should be prepared to address inquiries related to this domain. 

The counseling process should specifically focus on three key aspects. Firstly, it should discuss the potential benefits of screening in enhancing disease detection rates while acknowledging the inherent risks of false positive and false negative outcomes. Secondly, emphasis must be placed on the significance of regular screening tests, emphasizing the need for ongoing commitment rather than relying solely on a single baseline assessment. Thirdly, it should be emphasized that a positive result from a screening test does not establish a definitive diagnosis. Instead, further evaluation, including invasive procedures and biopsies, might be necessary to confirm the presence of the disease. 

In addition to these counseling points, recommendations aimed at reducing the risk of urothelial carcinoma development, such as lifestyle modifications like smoking cessation and avoiding exposure to known carcinogens in occupational settings, physical exercise, and improved diet, should be highlighted as well [143].

Moreover, the function of a biomarker in clinical decision making would exhibit distinctions between low/intermediate risk of NMIBC cases and high-risk NMIBC cases. Within individuals afflicted with low-grade disease, there is potential for a marker to decrease the frequency of required cystoscopies. Conversely, for high-grade cancers, the biomarker would serve as a supplementary tool alongside cystoscopy, where an abnormal outcome would increase awareness for both patients and physicians. This would help recognize individuals at a higher risk of progression, enhance the interpretation of inconclusive cytology results, and facilitate the evaluation of response to BCG treatment [144].

## 8. Discussion

FDA-approved urinary biomarkers for bladder cancer are prone to false positive results in 12–26% of patients without bladder cancer. Additionally, their limited sensitivity can lead to missed diagnoses in as high as 43% of patients with bladder cancer [87].

The pretest probability of a patient, which is the likelihood that they have bladder cancer based on certain factors like age, symptoms, environment, and other risk factors, should be considered when interpreting the results of urinary biomarkers. Patients with an elevated pretest probability are prone to receiving a positive test result, even if the biomarker’s sensitivity is relatively low. It is also crucial to consider the patient’s medical background and additional examinations, including cystoscopy and cytology. These assessments play a vital role in verifying or excluding the diagnosis of bladder cancer [65].

The potential benefit of urinary biomarkers depends on the situation in which they are employed. For instance, a urinary biomarker used as a diagnostic tool in a patient with hematuria will require a high negative predictive value and high specificity. Patients with hematuria should be categorized by gross and microscopic hematuria, with the former receiving cystoscopy. For patients with only microscopic hematuria, urinary markers can be an essential adjunct to nomograms leading to a more accurate evaluation of their disease status [145].

In high-grade stage tumors, urologists are unlikely to rely solely on biomarkers alone or in combination. Instead, they are more inclined to work with conventional approaches like cystoscopy and cytology. However, in cases of these advanced tumors, urinary biomarkers offer additional benefits like assessing the tumor’s aggressiveness, aiding in planning treatment intensification, and closer follow-ups [76]. A pioneering study by Todenhöfer et al. [136] shows the potential of combining urinary markers to predict aggressiveness. Their findings revealed that the co-presence of positive urine cytology and NMP22 indicated a significantly elevated risk (20-fold) of G3/CIS tumors, demonstrating the potential of urinary biomarkers to provide valuable insights into tumor behavior for better-informed medical decisions.

Undoubtedly, bladder cancer has several features that make it suitable for screening in the absence of symptoms. These characteristics comprise increased prevalence in high-risk individuals, well-established risk factors, well-known risk factors, the potential for a more favorable outcome when the disease is detected early, and ease of access to a source of disease markers, in this case, urine. Cystoscopy, the current gold standard procedure for diagnosing urothelial carcinoma, is highly sensitive and specific but does not lend itself to widespread use in screening programs. Urine cytology is useful and remains the current standard for detecting high-grade tumors. However, its low sensitivity—particularly for low-grade disease—limits its potential for sole use in screening. Previously, screening protocols were hampered by low disease prevalence, but introducing risk calculators may represent an acceptable cost/benefit ratio in moderate and high-risk individuals and microhematuria may be a useful initial screening modality for applying this risk stratification [63]. However, its beneficial potential is still limited by the fact that it can still be caused by many noncancerous conditions, resulting in low positive predictive value. As discussed above, developing tests and marker panels with improved performance metrics, and applying these in combination with risk stratification, is an approach that should be tested and validated in high-quality trials. Such a strategy may hold the key to successful screening and could also help to reduce the need for frequent cystoscopy in high-risk and surveillance populations to detect new and recurrent cases of urothelial carcinoma.

The FDA-approved biomarkers almost uniformly suffer from high false favorable rates due to benign inflammatory conditions. Urinary biomarkers may yield false positive results in 12%−26% of patients without bladder cancer. This is coupled with its limited sensitivity when used in isolation, leading to a missed diagnosis in up to 43% of patients with bladder cancer [104]. Although molecular bladder cancer assays have shown superior sensitivity compared to urine cytology, none have been included in clinical guidelines, as this low specificity remains one of the most significant limitations [76].

A wide range of promising markers for bladder cancer are currently under investigation, including protein- and gene-related markers, DNA methylation, miRNA, microsatellite analysis, and extracellular vesicles and exosomes, among other potential sources of biomarkers [146]. In the context of protein markers, some notable members include the following: UBC, CYFRA21-1, BLCA-4, CellDetect, hyaluronic acid, sFas, survivin, MCM%, ADXBladder, and URO17, among others. In this category, the protein marker URO17 is notable for its ability to identify Keratin 17, an oncoprotein crucial in cancer cell replication. The detection method employed is immunocytochemistry. Remarkably, this marker has consistently demonstrated high sensitivity and specificity in three separate studies. On average, it achieves sensitivity of 98% and specificity of 97% [147,148,149]. 

Gene-related biomarkers encompass a variety of candidates, including FGFR3, TERT, OTX1, HS3ST2, SEPTIN0, SLIT, FGFR3, SOX-1, IRAK3, Li-MET, histone tail modifications H3K9 and H3K27, miRs, miRNAs, Xpert Bladder, BlaDimiR, AssureMDX, CXBladder, Epicheck, Uromonitor, UroSEEK, and microsatellite analysis [146]. 

Numerous tests have been conducted to identify mRNA biomarkers and multi-gene panels. Among these, we are going to mention a few: CxBladder has undergone extensive examination, with different variations serving distinct purposes: Cxbladder^®^ Detect is employed to detect bladder cancer in hematuria patients, demonstrating sensitivity of 82% and specificity of 85% [150]. Cxbladder^®^ Triage is used for hematuria patients to rule out bladder cancer, achieving a negative predictive value of 97% and sensitivity of 95% [151]. Cxbladder^®^ Monitor serves as a complement to surveillance and has been compared with urine cytology, NMP22 BladderChek, and NMP22 ELISA. It exhibits significantly higher sensitivity and specificity of 91/96 compared to 22/87%, 11/87%, and 26/86%, respectively [152]. CxBladder Monitor (CxBM) was incorporated into local guidelines, where low-risk patients alternated between annual CxBM and cystoscopy thereafter [153]. Their findings showed that 77.8% of patients could safely manage with just one cystoscopy every 2 years, leading to a 39% reduction in the total number of annual cystoscopies. This practical advantage of CxBM in clinical practice has contributed to its increased utilization [154].

Within this gene-related biomarker category, certain biomarkers have gained attention due to their superior sensitivity and specificity in prospective studies. SOX-1, IRAK3, and Li-MET are characterized by detecting changes in DNA methylation within bladder cancer cells shed in urine, achieving sensitivity of 86% and specificity of 89%. They exhibit an 80% prediction rate for tumor recurrence, surpassing the performance of cytology (35%) and cystoscopy (15%) [155]. Also in this category, AssureMDX boasts sensitivity of 97% and specificity of 83% [156,157]. 

Uromonitor, another gene-related biomarker, is capable of detecting minute quantities of TERT promoter and FGFR3 hotspot mutations, common somatic alterations in bladder cancer [158]. An ongoing multicenter, observational, prospective study involving 146 patients is currently underway. All patients with low-grade NMIBC diagnosed within the last three years, either as a primary tumor or recurrence, were included in the study. Voided urine samples were collected immediately before the flexible cystoscopy. The Uromonitor test demonstrated sensitivity of 89.7% and specificity of 96.2%. The research aimed to assess the clinical utility of the Uromonitor test in accurately ruling out a recurrence when flexible cystoscopy yielded false positive results. Among the patients who had positive flexible cystoscopies and subsequently underwent transurethral resection of the bladder (TUR-B), histological examinations revealed 28 cases of false positives. Interestingly, the Uromonitor tests were also negative for all 28 of these patients. The results of the study suggest that using the Uromonitor test could have prevented 42% (28 out of 67) of unnecessary TUR-B procedures [159].

Lastly, microsatellite analysis (MSA) utilizes PCR to target highly variable short tandem repeats (STRs) that occur in cancer cells, exhibiting loss of heterozygosity (LOH), a result of epigenetic silencing or mismatch repair gene inactivation integral to cancer cell proliferation. In comparison to urine cytology, MSA demonstrates sensitivity of 97% versus 79%, particularly excelling in low-grade tumors (95–100%) based on a small study [160]. A separate prospective study of 91 patients evaluating MSA in combination with cytology revealed sensitivity of 72% for G1–2 tumors and 96% for G3 tumors. Additionally, the use of LOH analysis improved specificity, successfully identifying all recurrence cases [161]. Overall, MSA exhibits greater sensitivity; however, larger prospective studies with a validation cohort are necessary to fully assess its feasibility [146].

Given that the total cost of cystoscopy amounts to approximately USD 216.18 per procedure, and given its invasive nature, it becomes crucial for the benefits of early detection to outweigh the potential issues associated with unnecessary cystoscopy. These issues include discomfort, bleeding, infection, urethral trauma, and increased anxiety for patients. While early treatment of recurrences can prevent disease progression and future relapses, it is doubtful that a minor difference in detection time holds significant clinical relevance [13]. In this context, more affordable, non-invasive urinary biomarkers could potentially play a key role in reducing both financial burdens and patient morbidity while still maintaining a sensitive and specific diagnostic approach to reduce cancer progression and complications.

## 9. Conclusions

The clinical significance of a biomarker in guiding decisions related to bladder cancer management depends on several factors, including the patient’s initial risk profile, histology findings, and whether they have low- or high-risk non-muscle or muscle-invasive cancer. In cases where patients have low-grade disease, incorporating a biomarker could potentially reduce the necessity for frequent cystoscopies. Conversely, for individuals with high-grade cancers, the biomarker would complement cystoscopy as a diagnostic tool. Looking ahead, urinary biomarkers will play a key role in identifying those at risk of disease progression, aiding in the interpretation of inconclusive cytology results and assessing the response to BCG treatment and tumor recurrence prediction.

Additionally, it is imperative to acknowledge the current limitations of commercial urinary biomarkers in clinical settings, as their practical significance has not been definitively established. Nevertheless, numerous novel biomarkers have emerged and are presently undergoing trials, and the resurging enthusiasm for biomarkers is clear. After the execution of prospective studies and meta-analyses, the coming years are likely to witness the integration of a new generation of biomarkers into clinical practice, potentially resulting in positive outcomes in terms of morbidity and mortality.

## Figures and Tables

**Table 1 ijms-24-14374-t001:** Characteristics of FDA-approved urinary bladder cancer markers according to current guidelines.

FDA-Approved Urinary Markers	Sensitivity	Specificity	PPV	NPV	Actual Role in Clinical Guidelines
BTA Stat (POC)	40–72	29–96	40–88	38–76.9	--------
BTA TRAK	50–62	68–87	45.4	88.4	--------
NMP22^®^ BladderChek^®^ (POC)	11–85.7	77–100	18.2–100	61.9–93.9	--------
NMP22^®^ Bladder Cancer Test	24–81	49–100	31–100	60–91	---------
UroVysion^®^	13–100	63–100	21–83	67.9–100	Might serve as a reflex test following unremarkable cystoscopy findings and inconclusive or ambiguous cytology results
ImmunoCyt^®^	50–85	62–86	26–72	81–93	Might serve as a reflex test following unremarkable cystoscopy findings and inconclusive or ambiguous cytology results

Adapted from up-to-date catalog of available urinary biomarkers for the surveillance of non-muscle invasive bladder cancer by Soria F, Droller MJ, Lotan Y, Gontero P, D’Andrea D, Gust KM, Rouprêt M, Babjuk M, Palou J, Shariat SF. 2018., World Journal of Urology [75] Abbreviations: POC, point of care; PPV, positive predictive value; NPV, negative predictive value.

**Table 2 ijms-24-14374-t002:** Pooled sensitivities of FDA-approved urinary bladder cancer markers from meta-analysis.

Biomarker	Studies, n	Total Sample (TP), n	Sensitivity (95% CI)
Quantitative NMP22			
Overall	19	2002 (1237)	0.69 (0.62–0.75)
Evaluation of symptoms	9	368 (235)	0.67 (0.55–0.77)
Surveillance	10	1410 (832)	0.61 (0.49–0.71)
Qualitative NMP22			
Overall	4	304 (168)	0.58 (0.39–0.75)
Evaluation of symptoms	2	145 (69)	0.47 (0.33–0.61)
Surveillance	2	159 (99)	0.70 (0.40–0.89)
Qualitative BTA			
Overall	22	1403 (894)	0.64 (0.58–0.69)
Evaluation of symptoms	8	372 (275)	0.76 (0.67–0.83)
Surveillance	11	544 (325)	0.60 (0.55–0.65)
Quantitative BTA			
Overall	4	186 (125)	0.65 (0.54–0.75)
Evaluation of symptoms	1	49 (37)	0.76 (0.61–0.87)
Surveillance	2	67 (39)	0.58 (0.46–0.69)
FISH			
Overall	11	633 (416)	0.63 (0.50–0.75)
Evaluation of symptoms	2	144 (82)	0.73 (0.50 –0.88)
Surveillance	7	299 (189)	0.55 (0.36–0.72)
ImmunoCyt			
Overall	14	1042 (810)	0.78 (0.68–0.85)
Evaluation of symptoms	6	401 (334)	0.85 (0.78–0.90)
Surveillance	7	406 (302)	0.75 (0.64–0.83)

Adapted from Urinary Biomarkers for Diagnosis of Bladder Cancer: A Systematic Review and Meta-analysis by Chou R, Gore JL, Buckley D, Fu R, Gustafson K, Griffin JC, Grusing S, Selph S. 2015. Annals of internal medicine [87].

**Table 3 ijms-24-14374-t003:** Pooled specificities of FDA-approved urinary bladder cancer markers from meta-analysis.

Biomarker	Studies, n	Total Sample (TN), n	Specificity (95% CI)
Quantitative NMP22			
Overall	19	4472 (3555)	0.77 (0.70–0.83)
Evaluation of symptoms	7	945 (798)	0.84 (0.75–0.90)
Surveillance	8	2398 (1859)	0.71 (0.60–0.81)
Qualitative NMP22			
Overall	4	2325 (2039)	0.88 (0.78–0.94)
Evaluation of symptoms	2	1671 (1477)	0.93 (0.81–0.97)
Surveillance	2	654 (562)	0.83 (0.75–0.89)
Qualitative BTA			
Overall	21	2730 (2108)	0.77 (0.73–0.81)
Evaluation of symptoms	6	649 (526)	0.78 (0.66–0.87)
Surveillance	8	1003 (771)	0.76 (0.69–0.83)
Quantitative BTA			
Overall	4	246 (180)	0.74 (0.64–0.82)
Evaluation of symptoms	1	47 (25)	0.53 (0.38–0.68)
Surveillance	2	131 (104)	0.79 (0.72–0.85)
FISH			
Overall	11	1188 (1034)	0.87 (0.79–0.93)
Evaluation of symptoms	2	507 (481)	0.95 (0.87–0.98)
Surveillance	6	468 (361)	0.80 (0.66–0.89)
ImmunoCyt			
Overall	14	3445 (2656)	0.78 (0.72–0.82)
Evaluation of symptoms	7	1475 (1257)	0.83 (0.77–0.87)
Surveillance	8	1079 (823)	0.76 (0.70–0.81)

Adapted from Urinary Biomarkers for Diagnosis of Bladder Cancer: A Systematic Review and Meta-analysis by Chou R, Gore JL, Buckley D, Fu R, Gustafson K, Griffin JC, Grusing S, Selph S. 2015. Annals of internal medicine [87].

## Data Availability

The datasets generated and/or analyzed during the current study are publicly available.

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
