# Peer review of "Molecular Markers for Bladder Cancer Screening: An Insight into Bladder Cancer and FDA-Approved Biomarkers"

_ijms, 2023, doi:10.3390/ijms241814374_

Round 1

Reviewer 1 Report

General comment

The manuscript entitled “Molecular Markers for Bladder Cancer Screening: An Insight From 4 FDA-Approved Tests” by Flores Monar et al., aims to summarize, in a comprehensive review, the characteristics of FDA-approved assays utilized in the diagnosis and follow-up of BC. Overall, the manuscript is similar to other papers already published in the literature, however, if properly revised, considering the novel and latest data about the urinary biomarkers in BC, it could be a work enriching the current literature. Additionally, considering that the main point of the paper, according to the abstract and the title is to report the role and the characteristics of four FDA-approved biomarkers, I would expect a proper discussion on available data. Other corrections are suggested in the following list.

TITLE

I would avoid reporting the fact that there are 4 FDA tests. If you want to do it, at least write numbers in letters.

INTRODUCTION

Regarding the epidemiology of BC, also see DOI: 10.2144/fsoa-2020-0210. And doi: 10.7759/cureus.27330

51-53: The role of enhanced cystoscopy based on fluorescence or narrow-band imaging is however utilized in recurrence settings and should be properly explained. It would be better to move this information in a context related to BC monitoring or therapy (enhanced TURBT). A proper reference is also needed.

63: check typos

BLADDER CANCER OVERVIEW

114: In addition to the most known risk factors, a nice addition would be to report emergent risk factors for BC. See DOI: 10.1007/s10552-023-01711-0, DOI: 10.3390/jpm13030512, DOI: 10.1007/s00125-006-0468-0 and DOI: 10.3390/cancers14194775

145: Probably, it would be better to report these subparagraphs in another paragraph. Additionally, the transition to the epidemiologic data and the existing non-invasive method for the assessment of BC is not particularly flowing.

163: Similar to before, if you report urine cytology, considering the common and standard method, you should highlight the novel urinary biomarkers that are conceptually and methodologically different.

190-403: all this section is quite similar to other articles already published in the literature. I suggest you to analyze these papers and properly cite them in your work. Additionally, provide the latest data for the reported biomarkers in order to avoid the redundancy of information with those papers.

493: considering the length of the paper, the use of only 3-4 paragraphs it would probably not be enough in order to provide a tidy and clean paper. Considering to destructure the second paragraph into two or three smaller ones.

DISCUSSION

533: In addition to the interesting paragraph related to the discussion of the use of novel urinary biomarkers with the patients, it should also be noted that these novel biomarkers have different limitations that should be reported. Lastly, the analysis of costs related to the use of these biomarkers and the standardized diagnosis (cystoscopy mainly) could be a nice addition and point of discussion.

CONCLUSION

599: Provide future perspectives about the use of urinary biomarkers.

FIGURE

The only figure reported is not particularly informative and should be accompanied by other figures or tables in order to provide a better background for your study.

TABLES

A table reporting the ongoing clinical trials would also be a nice addition.

minor grammar errors and typos

Author Response

REVIEW

"Suggested recommendations are marked with green highlighting, while corrections and their corresponding reviews are indicated in yellow. The corrected article also features yellow highlights for these changes."

1.Suggested: TITLE. I would avoid reporting the fact that there are 4 FDA tests. If you want to do it, at least write numbers in letters.

Reviewed: Title is now: Molecular Markers for Bladder Cancer Screening: An Insight from Bladder Cancer and FDA-Approved Biomarkers

  1. Suggested: INTRODUCTION. Regarding the epidemiology of BC, also see DOI: 10.2144/fsoa-2020-0210. And doi: 10.7759/cureus.27330

Reviewed: We have thoroughly reviewed the suggested articles and have integrated pertinent information to enrich the epidemiological content. Page 1. (Lines 33-42)

  1. Suggested: 51-53: The role of enhanced cystoscopy based on fluorescence or narrow-band imaging is however utilized in recurrence settings and should be properly explained. It would be better to move this information in a context related to BC monitoring or therapy (enhanced TURBT). A proper reference is also needed.

Reviewed: We have added a new section focused on the diagnosis of the condition, which includes information about enhanced cystoscopy along with its corresponding references.  Pages 6,7 (Lines 302-314)

  1. Suggested: BLADDER CANCER OVERVIEW. 114: In addition to the most known risk factors, a nice addition would be to report emergent risk factors for BC.

See DOI: 10.1007/s10552-023-01711-0. We added risk factors for bladder cancer associated to physical activity and sitting time based on this article. Pages 4,5. (Lines 201-209)

DOI: 10.3390/jpm13030512. We added the relationship between cardiovascular disease and bladder cancer based in this retrospective analysis. Page 4. (Lines 161-172)

DOI: 10.3390/cancers14194775. Bladder cancer and red meat consumption correlation was addressed based on this article. Page 5. (Lines 212-220)

  1. Suggested: 145: Probably, it would be better to report these subparagraphs in another paragraph. Additionally, the transition to the epidemiologic data and the existing non-invasive method for the assessment of BC is not particularly flowing.

Reviewed: Now after risks factors of bladder cancer comes presentation and diagnosis and the transition flows more properl Page 6. (Line 265).

  1. Suggested:163: Similar to before, if you report urine cytology, considering the common and standard method, you should highlight the novel urinary biomarkers that are conceptually and methodologically different.

Reviewed: Cytology is compared to the biomarkers mentioned in the article with its respective sensitivities and specificities.

  1. Suggested. 190-403: all this section is quite similar to other articles already published in the literature. I suggest you to analyze these papers and properly cite them in your work. Additionally, provide the latest data for the reported biomarkers in order to avoid the redundancy of information with those papers.

Reviewed: Information has been updated

  1. Suggested. 493: considering the length of the paper, the use of only 3-4 paragraphs it would probably not be enough in order to provide a tidy and clean paper. Considering to destructure the second paragraph into two or three smaller ones.

Reviewed: Paragraph structure has been improved. Page 15. (Line 652)

  1. Suggested. 533: In addition to the interesting paragraph related to the discussion of the use of novel urinary biomarkers with the patients, it should also be noted that these novel biomarkers have different limitations that should be reported.

Reviewed: Limitations of the urinary biomarkers have been reported in the article

  1. Suggested. Lastly, the analysis of costs related to the use of these biomarkers and the standardized diagnosis (cystoscopy mainly) could be a nice addition and point of discussion.

Reviewed: Cystoscopy costs have been added in the discussion. Page 2. (Line 68)

  1. Suggested. CONCLUSION. 599: Provide future perspectives about the use of urinary biomarkers.

Reviewed: Future perspectives in conclusion have been added. Page 17. (Line 808)

  1. Suggested. FIGURE. The only figure reported is not particularly informative and should be accompanied by other figures or tables in order to provide a better background for your study.

Figure was deleted and 2 additional tables were added about sensitivities and specificities comprising data from meta-analysis from FDA approved biomarkers. (Page 17)

Reviewer 2 Report

To the authors of the manuscript "Molecular markers for bladder cancer screening: An insight from 4 FDA-Approve Tests".

The following are my recommendations:

-          Lines 105-112. It does not talk about age as a risk factor but about incidence. Rewrite paragraph.

-          If the first risk factor considered is tobacco, please put it first and elaborate a little more. Regarding exposure to substances, it should be remembered that many of the jobs are accessible to men, which has a direct implication in the 3:1 ratio in men, in addition to mention that genetic polymorphisms affecting enzymes involved in the activation of amines can determine susceptibility to the effects of such substances (Burger 2013,pelucchi 2006). In addition single nucleotide polymorphisms (SNPs) have been associated with a significantly increased risk in bladder cancer in GWAS studies (Dobruch 2016). Most of the significant variants associated with bladder cancer risk are found in genes pertaining to carcinogen metabolism by xenobiotic enzymes and it should be noted that although the characterization of genetic susceptibility to this cancer has provided significant findings, it is still limited because it is necessary to apply multiple marker analysis against SNPs due to the heterogeneity of tumors (Maturana 2018,Babjuk 2017).

-          Differences regarding the underlying anatomy between men and women, as well as variations in hormone receptors could play a key role, as there are different studies that support the idea that the effect of androgens or estrogens on bladder carcinogenesis and hepatic metabolism may contribute to gender differences (Patel 2015, Dobruch 2015, Malats and Real 2015, Antoni 2017...). )For example, altered androgen levels in women facilitate the early activation of pathways that stimulate tumor growth and metastasis which may contribute to a more aggressive tumor biology (Gakis and Stezl 2013).

-          Lines 131-136.  References?

-          Regard to diagnosis and hematuria, it should be remembered that women suffer more delays in diagnosis. That 70% of patients have benign hematuria due to stones, urine infection, vaginal bleeding in women or benign prostatic hyperplasia in men and that all this makes diagnosis more difficult.

-          Please make reference to the European urology guidelines (Babjuk 2017,Witjes 2017).

-          CXBaldder is not commented. Combines genetic analysis with clinical diagnostic data. It has a sensitivity of 91% (higher than cystoscopy) and is used in patients at high risk of recurrence. It addresses five specific mRNAs and does not require FDA approval (Duquesne 2017, O'Sullivan 2012).

-          Current role in clinical guidelines (table 1): BTAs are for surveillance, not diagnosis; NMP22 (POC) follow-up of NMIBC; NMP22 bladder cancer test for initial high-grade diagnosis.

-          Line 449-452: it is important to note the lack of a consensus and reproducible grading system that reflects the different diagnostic criteria adopted by practitioners. the grade classification not only includes WHO1998-2004, but also staging. In addition with the 2004 updates the TNM is also updated to include nodal invasion and metastatic grade, so it is not only for staging. There is an update to the WHO classification published in 2016 that they should review (Humprey 2016).

-          Although a review is not an original research article, it should have an objective that is original, it should not be a mere collection of data that many other references have. For example, it would be very interesting to talk not only about what is approved by the FDA, if not those tests that are on the way and that are highly promising or those that do not have approval ,because you do not have it?. Here are two very interesting examples:

1. BladimiR         https://pubmed.ncbi.nlm.nih.gov/36085102/

2.Uromonitor    https://uromonitor.com/

-          In addition, it would be important to talk a little more about the outlook for liquid biopsy markers and their approval by the world agencies. In the conclusions you comment that there are numerous new biomarkers that are emerging, could you comment on some of them? Not in conclusions, better in discussion.

-          Finally, I leave a clinical trial very updated and approved by the FDA that would be relevant to comment: https://classic.clinicaltrials.gov/ct2/show/NCT03962933

-          From the 130 bibliographic references, 75% are more than 20 years old, please reduce the number of references by including reviews that encompass tests and guidelines and more updated references. Such as Burger 2013,, Cumberbatch 2018, Duquesne 2017, Babjuk 2017 amoung others.

Overall, I do not find anything original and novel in this article that is not in many others and it would be nice if they included novelties for it to have a greater impact. The authors need to do more research and go deeper in the field to reference in a more current way and with less references as well as in the latest trials or tests that are in development.

Author Response

Review 2

"Suggested recommendations are marked with green highlighting, while corrections and their corresponding reviews are indicated in yellow. The corrected article also features yellow highlights for these changes."

  1. Suggested: Lines 105-112. It does not talk about age as a risk factor but about incidence. Rewrite paragraph.

Reviewed:  I updated the content about risk factors Pages 3-6. (Lines 109-263) and the paragraph about age is now in the introduction as epidemiology and incidence Page 2. (Lines 53-60)

  1. Suggested: If the first risk factor considered is tobacco, please put it first and elaborate a little more.

      Reviewed: New content and references have been added about a deeper comprehension about tobacco as a risk factor Page 4. (Lines 111-131)

  1. Suggested: Regarding exposure to substances, it should be remembered that many of the jobs are accessible to men, which has a direct implication in the 3:1 ratio in men.

Susceptibility to develop bladder cancer according to occupational exposure and gender was mentioned in risk factors. Page 2. (Lines 152-158)

  1. Suggested: In addition to mention that genetic polymorphisms affecting enzymes involved in the activation of amines can determine susceptibility to the effects of such substances (Burger 2013, Pelucchi 2006).

Genetic susceptibility is now mentioned in risk factors Page 4. (Lines 175-198)

  1. Suggested: In addition single nucleotide polymorphisms (SNPs) have been associated with a significantly increased risk in bladder cancer in GWAS studies (Dobruch 2016). Most of the significant variants associated with bladder cancer risk are found in genes pertaining to carcinogen metabolism by xenobiotic enzymes and it should be noted that although the characterization of genetic susceptibility to this cancer has provided significant findings, it is still limited because it is necessary to apply multiple marker analysis against SNPs due to the heterogeneity of tumors (Maturana 2018, Babjuk 2017).

SNPs has been mentioned now in the article in the risks factors. Page 4. (Lines 175-198)

  1. Suggested: Differences regarding the underlying anatomy between men and women, as well as variations in hormone receptors could play a key role, as there are different studies that support the idea that the effect of androgens or estrogens on bladder carcinogenesis and hepatic metabolism may contribute to gender differences (Patel 2015, Dobruch 2015, Malats and Real 2015, Antoni 2017...). )For example, altered androgen levels in women facilitate the early activation of pathways that stimulate tumor growth and metastasis which may contribute to a more aggressive tumor biology (Gakis and Stezl 2013).

Reviewed: Androgens and estrogens influence have been addressed; gender wise. Page 5. (Lines 223-240)

  1. Suggested: Lines 131-136.  References?

 Reviewed: Updated information about occupational exposure have been added with its respective references. Pages 2,3 (Lines 134-158)

  1. Suggested: Regard to diagnosis and hematuria, it should be remembered that women suffer more delays in diagnosis. That 70% of patients have benign hematuria due to stones, urine infection, vaginal bleeding in women or benign prostatic hyperplasia in men and that all this makes diagnosis more difficult.

 Reviewed: Gender disparities and delay in diagnosis in women is now shown. Page 3,4. (Lines 241-263)

  1. Suggested: Please make reference to the European urology guidelines (Babjuk 2017,Witjes 2017).

Guidelines have been reviewed

  1. Suggested: CXBaldder is not commented. Combines genetic analysis with clinical diagnostic data. It has a sensitivity of 91% (higher than cystoscopy) and is used in patients at high risk of recurrence. It addresses five specific mRNAs and does not require FDA approval (Duquesne 2017, O'Sullivan 2012).

CXBladder is now explained in discussion Pages16,17 (752-766) 

  1. Suggested:Line 449-452: it is important to note the lack of a consensus and reproducible grading system that reflects the different diagnostic criteria adopted by practitioners. the grade classification not only includes WHO1998-2004, but also staging. In addition with the 2004 updates the TNM is also updated to include nodal invasion and metastatic grade, so it is not only for staging. There is an update to the WHO classification published in 2016 that they should review (Humprey 2016).

We deleted this part as it was not necessary or added value to the content of the review in reference to biomarkers

  1. Suggested: Although a review is not an original research article, it should have an objective that is original, it should not be a mere collection of data that many other references have. For example, it would be very interesting to talk not only about what is approved by the FDA, if not those tests that are on the way and that are highly promising or those that do not have approval ,because you do not have it?. Here are two very interesting examples. In addition, it would be important to talk a little more about the outlook for liquid biopsy markers and their approval by the world agencies. In the conclusions you comment that there are numerous new biomarkers that are emerging, could you comment on some of them? Not in conclusions, better in discussion.

  Now, novel urinary biomarkers are mentioned in discussion

  1. Suggested: Finally, I leave a clinical trial very updated and approved by the FDA that would be relevant to comment: https://classic.clinicaltrials.gov/ct2/show/NCT03962933

The ongoing trial has been mentioned now in the article. Page 17. (Lines 774-787)

  1. Suggested: From the 130 bibliographic references, 75% are more than 20 years old, please reduce the number of references by including reviews that encompass tests and guidelines and more updated references. Such as Burger 2013,, Cumberbatch 2018, Duquesne 2017, Babjuk 2017 amoung others.

Between 15- 20% of references have more than 20 years but some of them are historically necessary to address as most FDA approved biomarkers have been developed several years ago.

Round 2

Reviewer 1 Report

The authors improved the manuscript accordingly. No further corrections required.

Minor